# Novel method for combining microbial bioremediation with static magnetic fields to remediate mercury-contaminated soils

Naima Werfelli[1,2], Mariem Taboubi[1], Sirine Ridene[1], Hadir Bousselmi[1], Ahlem Mansouri[1], Ahmed Landoulsi[1], Chiraz Abbes [1]*

**1** Biochemistry and Molecular Biology Laboratory, Risks Related to Environmental Stress, Struggle and Prevention (UR17ES20), Faculty of Sciences of Bizerte University of Carthage, Zarzouna Bizerte, Tunisia, **2** International Center for Environmental Technologies (CITET), Boulevard Leader Yasser Arafat, Tunis, Tunisia

* chiraz.abbes@gmail.com

## Abstract

Heavy metal contamination poses a significant risk to both the environment and public health, particularly through metallic mercury, a neurotoxic contaminant capable of bioaccumulating in food chains. This article presents a novel approach to remediating mercury-polluted soils by combining microbial bioremediation with the effects of a static magnetic field, applied at an induction of 260 mT for 12 hours at the start of the experiment. The decontamination technique was applied to mercury-contaminated soil bioaugmented with the bacterial strain *Pseudomonas stutzeri* LBR. Mercury remediation was enhanced by the static magnetic field in conjunction with bioaugmentation over a 30-day period. Notably, in non-sterile soils, the combination of an SMF, total soil flora, and *Pseudomonas stutzeri* LBR increased mercury remediation efficiency by 49.36%, compared to only 23.85% in the absence of an static magnetic field and soil bioaugmentation. Similarly, in sterile soils, the combination of an static magnetic field and *Pseudomonas stutzeri* LBR increased mercury remediation efficiency by 72.49%, compared to 38.1% without an static magnetic field and soil bioaugmentation. This study highlights the potential of combining an static magnetic field with microbial bioremediation to accelerate the remediation of mercury-contaminated soils, suggesting that this approach may become increasingly important in the future.

## Introduction

Mercury (Hg), a major environmental pollutant, is a persistent, bioaccumulative toxic metal. Notably, Hg is ranked third by the U.S. Agency for Toxic Substances and Disease Registry (ATSDR) because of its high abundance, significant toxicity, and considerable potential for human exposure [1]. Methylmercury (MeHg), one of its most concerning forms, is classified as a possible human carcinogen (Group 2B)

**Data availability statement:** All relevant data are within the manuscript and its Supporting Information files.

**Funding:** This research was supported by institutional funding from the Ministry of Higher Education and Scientific Research of Tunisia. The funders had no role in study design, data collection and analysis, decision to publish, or preparation of the manuscript.

**Competing interests:** The authors have declared that no competing interests exist.

by the International Agency for Research on Cancer (IARC) [2]. Mercury originates from both natural sources, such as volcanic eruptions and weathering of rocks, and anthropogenic activities, including coal combustion and mining [3]. Its global dispersion and bioaccumulation pose risks to the environment as well as animal and human health [4,5].

Mercury is present in various environments, including terrestrial, marine, and atmospheric systems, in multiple chemical forms that influence its mobility, persistence, and toxicity [6]. The predominant form in the atmosphere is gaseous elemental mercury ($Hg^0$), which accounts for the majority of mercury emissions and can travel long distances before deposition. In addition, oxidized mercury species, primarily divalent mercury ($Hg^{2+}$), are also present and more readily deposited in aquatic and terrestrial environments [7]. In aquatic systems, inorganic mercury ($Hg^{2+}$) undergoes microbial transformation into MeHg through biogeochemical processes involving sulfate- and iron-reducing bacteria [8,9]. The most toxic and bioavailable form, MeHg accumulates in aquatic organisms and magnifies throughout the food web, ultimately posing significant risks to human and wildlife health [10,11]. Among terrestrial ecosystems, soils represent the largest reservoir of mercury, where it predominantly exists in bound inorganic and organic forms [12]. While a minor fraction of mercury is re-emitted into the atmosphere as $Hg^0$ through volatilization, most remains sequestered in the soil, where it can be mobilized by environmental changes such as land-use disturbance or changes in redox conditions [13]. This persistence in soil contributes to its entry into the food chain through bioaccumulation in plants intended for human or animal consumption [14,15]. Since rice is a staple food for many populations worldwide, it is concerning that it bioaccumulates significant amounts of MeHg [16,17].

Although mercury and its compounds are toxic to all living organisms, some bacteria have been observed to possess resistance genes to mercury, indicating the potential for evolutionary adaptation to this pollutant [18,19]. A variety of bacterial species can resist toxic forms of mercury, such as methylmercury ($CH_3Hg^+$) and divalent mercury ($Hg^{2+}$), and subsequently convert them into nontoxic forms, such as elemental mercury ($Hg^0$) or insoluble compounds. The prevalence of mercury-resistant bacteria is directly proportional to the level of mercury contamination in the environment [12,20]. One of the most intriguing aspects of mercury resistance is the mechanism regulated by the *mer* operon, a well-characterized genetic system responsible for mercury detoxification. This operon is typically born on plasmids and other mobile genetic elements and includes key genes such as *merA*, *merR*, *merT*, *merP*, and *merB*. The *merA* gene, which encodes the mercuric reductase enzyme, catalyzes the reduction of $Hg^{2+}$ to $Hg^0$, a less toxic and volatile form of mercury [18,21]. This conversion is crucial because it transforms a highly toxic and bioavailable form of mercury ($Hg^{2+}$) into a less toxic and volatile form ($Hg^0$), thereby reducing its environmental and biological impact [5].

The remediation of polluted areas to remove mercury poses a significant challenge to the advancement of environmental protection. A variety of methods based on

biosorbents or ion-exchange resins have been employed to remove mercury [22], yet these approaches are not environmentally friendly.

In recent years, bioremediation of contaminated environments has emerged as a promising alternative with a lesser negative impact on ecosystems [23]. The bacteria used in this process are mercury-resistant because they carry *mer* operon genes, which can reside on transposons, plasmids, or the bacterial chromosome, as observed in strains of *Pseudomonas aeruginosa* [24], *Bacillus cereus* [25], and *Pseudomonas stutzeri* [26]. Zheng et al. (2018) showed that the marine strain *P. stutzeri* 273 possesses the *merA* gene and other components of the *mer* operon, such as *merT* and *merP*, which are involved in the uptake and transport of $Hg^{2+}$ into the cell. These findings suggest that *P. stutzeri* 273 is resistant to 50 µM $Hg^{2+}$ and can remove up to 94% of $Hg^{2+}$ from the culture medium under laboratory conditions in artificial LB medium [27].

In this study, for the Hg bioremediation experiment, we chose to use the *P. stutzeri* LBR strain, which was isolated and selected in recent studies by Mansouri et al. (2019) [28]. *P. stutzeri* LBR was tested for the first time for Hg bioremediation, although it previously demonstrated remarkable efficiency in degrading a mixture of pollutants, including pesticides and hydrocarbons, while also exhibiting resistance to heavy metals like lead. Specifically, it was capable of degrading 87% of 1,1,1-trichloro 2,2-bis(p-chlorophenyl) ethane (DDT) and 83% of benzo(a)pyrene (BaP) in 30 days [28]. Moreover, when bioaugmented individually in non-sterile soil, it effectively reduced lead concentrations by 58.07% [29].

Our primary interest in this work was to apply a novel technology aimed at accelerating the microbial bioremediation of polluted soils, namely the application of a static magnetic field (SMF). SMFs have been widely used in wastewater bioremediation with activated sludge systems. Scientific studies demonstrate that SMF application enhances both wastewater treatment efficiency and organic pollutant biodegradation [30]. One study demonstrated that applying a 7.5 mT SMF to activated sludge systems enhanced wastewater remediation, showing a 25% greater reduction in chemical oxygen demand compared to the control [31]. Křiklavová et al. (2014) reported that the application of a 370 mT SMF enhanced *Rhodococcus erythropolis*'s phenol degradation rates by 34% [32].

Further work from our group revealed that the combination of *P. stutzeri* LBR bioremediation and an SMF was an effective method for the remediation of organic pollutants (BaP and DDT). Indeed, the induction of an SMF (200 mT) doubled the bioremediation rate [28]. It would be intriguing to examine this combination's potential for the remediation of heavy metal-contaminated sites.

This study aims to evaluate the effectiveness of combining an SMF with bacterial bioremediation by *P. stutzeri* LBR for decontaminating soil (sterile and non-sterile) contaminated with mercury. The research seeks to determine whether this combined approach enhances mercury removal efficiency compared to the individual bioremediation method.

## Materials and methods

### Soil sampling and preparation

The soil samples were collected in the Kasserine region, located in the west-central part of Tunisia, within an industrial zone (GPS: 35.168534, 8.817806). This site was chosen due to its historical activity of industrial waste disposal for more than two decades. A total of 30 soil samples were collected over an area of approximately 300 m² to ensure a representative analysis of the site. Soil samples were collected using the quincunx sampling method, specifically from a depth of 0–20 cm beneath the soil surface. The samples were then placed in sterile bottles, stored at 4°C, and transported to the laboratory at the Tunisian International Center for Environmental Technologies (CITET) in compliance with ISO 11464:2006 [33].

The mercury concentration in the sample soil (S1) indicated low mercury pollution. For this reason, we decided to artificially contaminate soil S1 using the following method: a mercury solution was prepared by dissolving 148 mg of mercury sulfate ($HgSO_4$) in 100 mL of distilled water, acidified with 10 mL of sulfuric acid ($H_2SO_4$), and adjusted to a pH of 7

(maintain $Hg^{2+}$ solubility and prevent precipitation [34]) to obtain a 4.54 mM mercury solution. The entire prepared mercury solution was thoroughly mixed with 1 kg of soil S1 from the sampling area to obtain contaminated soil with a mercury concentration of 90 mg/kg, referred to as soil S2. Then, 50 g of soil S2 was mixed with 412 g of soil S1 to obtain soil S3, which had an appropriate mercury concentration for experimentation [35,36]. The final concentration of soil S3 after artificial contamination is 10.15 mg Hg/Kg soil. All bioremediation experiments in this study were conducted using soil S3. The mercury solution was thoroughly and uniformly mixed into the soil to ensure even distribution.

### Soil analysis (physical, chemical, and heavy metal concentration)

The pH was determined using a SevenCompact™ pH meter S220, following the standard method described in ISO 10390:2021 [37]. The dry matter (DM) percentage and moisture percentage were determined according to the standard method ISO 11465:1993 [38]. A 20 g soil sample ($m_0$) was dried in a Binder oven at 105°C for 24 hours until a constant mass ($m_1$) was obtained. The dry matter percentage and moisture percentage (on a fresh weight basis) were calculated using the following formulas, respectively:

$$\% \text{ DM } = (m_1/m_0) \times 100; \ \% \text{ Moisture (fresh basis)} = ((m_0 - m_1)/m_0) \times 100$$

The electrical conductivity of the soil extract was measured using a calibrated conductometer inoLab® Cond 7310 at 20°C ± 1°C, in accordance with the standard method, ISO 7888:1994 [39]. The soil extract was prepared by mixing 10 g of soil with 90 mL of distilled water. After homogenization, the suspension was allowed to settle, and the conductivity of the supernatant was measured directly. The measured value was corrected to 25°C and expressed in microsiemens per centimeter (µS/cm). Salinity was determined following the same principle as electrical conductivity, using the same conductometer.

The concentration of heavy metals (cadmium, cobalt, copper, iron, lead, manganese, nickel, zinc, chromium) and major elements (calcium, magnesium, sodium, potassium, phosphorus) in the tested soil sample was analyzed through inductively coupled plasma optical emission spectrometry (ICP-OES), in conformity with the ISO/TS 16965:2013 standard [40]. The ICP-OES apparatus utilized was a PerkinElmer Optima 7300 DV, manufactured in Shelton, CT, USA; the results are expressed as mg/kg dry matter (mg/kg DM) (S1 Table).

The Milestone Direct Mercury Analyzer (DMA-80) is a device used for determining total mercury levels during the bioremediation process. It is fully compliant with both the US EPA Method 7473 and the ASTM Method D6722-01. Its limit of detection is 0.01 mg/kg, and its limit of quantification is 0.03 mg/kg.

### Bacterial strain

A bacterial strain *Pseudomonas stutzeri* LBR [28] was isolated from the sediments of the Bizerte lagoon in our Biochemical and Molecular Biology Laboratory, which is part of Bizerte University's Faculty of Sciences. The GenBank database has been updated with the 16S sequences of the bacterial strain *P. stutzeri* LBR (accession number GenBank KC157911). This strain was selected due to its robust capacity to decompose a diverse range of organic contaminants, as well as its tolerance to heavy metals and its utility in bioremediation through bioaugmentation of lead-contaminated soils [29].

### Culture media

Luria–Bertani (LB; Difco, USA) medium consisted of 10 g NaCl, 10 g peptone, and 5 g yeast extract, with or without 20 g agar, in 1,000 mL distilled water, and the solution's pH was adjusted to 7. M9 medium contained (per liter) 200 mL M9 salt (12.8 g $Na_2HPO_4 \cdot 7H_2O$, 3 g $KH_2PO_4$, 0.5 g NaCl, 1 g $NH_4Cl$, 200 mL distilled water), 2 mL $MgSO_4$ 1 M, 0.1 mL $CaCl_2$ 1 M, and 2 g of sodium pyruvate. The solution's pH was adjusted to 7.

## Bioaugmentation tests and monitoring

We prepared four independent overnight precultures of *P. stutzeri* LBR in 10 mL of LB medium at 30°C, with shaking at 100 rpm. The following morning, the bacterial pellets were washed three times with M9 medium. Each wash was followed by centrifugation at 5,000 rpm for 5 minutes at 4°C. The final pellets were resuspended in 50 mL of sterile M9 medium to reach an optical density at 600 nm (OD600) of 0.1, corresponding to approximately $7 \times 10^9$ cells.

The bioaugmentation tests were conducted in four flasks. Each flask contained 20 g of soil blended with 40 mL of M9 medium, sterilized in accordance with standard protocols, and inoculated with 50 mL of bacterial pellet, prepared as described above. The experiment was conducted using four sterile flasks, each containing a different combination of soil supplemented with mercury (Hg), M9 medium with pyruvate, and *P. stutzeri* LBR, as follows: Flask 1: Non-sterile soil supplemented with Hg, M9 (pyruvate), and *P. stutzeri* LBR. Flask 2: Sterile soil supplemented with Hg, M9 (pyruvate), and *P. stutzeri* LBR. Flask 3: Non-sterile soil supplemented with Hg and M9 (pyruvate) without *P. stutzeri* LBR. Flask 4: Sterile soil supplemented with Hg and M9 (pyruvate) without *P. stutzeri* LBR. All containers were incubated at 30°C under aerobic conditions with agitation at 100 rpm for 30 days. Throughout the experiment, mercury concentration and *P. stutzeri* LBR counts were monitored every 15 days. To ensure reproducibility, three temporal replicates were conducted for each treatment, meaning that the entire experiment was repeated three times under identical conditions. Bacterial counts and mercury analyses were performed for each replicate at 15-day intervals. Strict aseptic conditions, including the use of sterilized materials (e.g., autoclaved glassware, sterile pipettes, and gloves) and procedures conducted under a laminar flow hood, were maintained throughout the study to prevent contamination.

## Static magnetic field (SMF) application

The static magnetic field exposure setup was developed at the Laboratory of Biochemistry and Molecular Biology, Faculty of Sciences, Bizerte. The apparatus comprises two cylindrical coils, separated by a distance of 11 cm, with a diameter of 20 cm and a thickness of 13 cm. Additionally, it includes an electromagnet (Beaudouin) and a direct current generator (frequency = 50 Hz). Additionally, the apparatus incorporates a field induction adjustment system, a coil cooling system (water pump), and an inoculum incubation assembly comprising a resistance and a pump that facilitates the circulation of water at 30°C through a double bottle system with an intermediate envelope (a space between the inner and outer bottles). The double bottle was constructed by a glassblower, and a digital tesla meter was employed to conduct regular field induction assessments [28,41,42] (S1 Fig).

After preparing the four treatments, the content of the flasks (soil supplemented with Hg and M9 (pyruvate), with or without *P. stutzeri* LBR) was then transferred into a double-walled glass vial and subjected to a magnetic field of 260 mT for 12 hours. Subsequently, the four flasks were placed in a shaking water bath at 30°C for 30 days to monitor Hg concentration and bacterial enumeration, as described above.

## Statistical analysis

The data were analyzed using STATISTICA 8.0 to calculate the means and standard deviations. All analyses were repeated in triplicate. A comparative analysis was conducted using an analysis of variance (ANOVA) test, followed by the post hoc Tukey test, to detect significant differences between groups, with statistical significance set at $p < 0.05$. The statistical software used was R Version 3.6.3 (R Development Core Team, 2009–2018, RStudio, Inc.).

## Results

### Soil analysis (physical, chemical, and heavy metal concentration)

The results of physicochemical characteristics in soil S1 demonstrated an alkaline pH of 9.23, a dry matter percentage of 85%, and a soil moisture percentage of 15%. Additionally, the soil exhibited a markedly low electrical conductivity value of 165.0 µS/cm and a negligible salinity value.

The concentration of heavy metals and major elements in soil S1 is presented in Table 1.

**Table 1. Elemental and heavy metal concentrations in soil S1 before artificial contamination by mercury.**

| Elements | Concentration (mg/kg DM) |
|---|---|
| Calcium | 38.47 |
| Magnesium | 27.05 |
| Sodium | 57.15 |
| Potassium | 74.30 |
| Phosphorus | 20.71 |
| Mercury | 0.3 |
| Cadmium | 0.0 |
| Cobalt | 0.0 |
| Copper | 39.13 |
| Iron | 6504 |
| Lead | 25.33 |
| Manganese | 320.2 |
| Nickel | 85.36 |
| Zinc | 68.66 |
| Chromium | 341.03 |

DM = Dry matter

The mercury concentration in soil S1 in its natural state was found to be relatively low, at approximately 0.3 mg/kg DM. In fact, natural mercury concentrations in uncontaminated soils typically range from 0.003 and 0.5 mg/kg DM [35]. In residential soils, intervention thresholds, although they vary across national regulations, generally converge around values between 7 and 10 mg/kg DM [36,43]. These findings indicate that soil S1 was not contaminated with mercury. Therefore, we artificially added mercury to achieve adequate concentrations for bioremediation.

Following artificial contamination, the mercury concentration in soil S3 reached toxic and polluting levels, with 10.15 mg/kg in non-sterile soil and 7.56 mg/kg in sterile soil. For the remainder of the experiments, this spiked soil S3 was used as the sample soil.

## Impact of SMF on bioaugmentation tests and monitoring

The results of monitoring the rate of mercury reduction in non-sterile soil not bioaugmented by *P. stutzeri* LBR in the presence and absence of an SMF are presented in Fig 1. After 30 days of the experiment at 30°C, we observed a decrease in the mercury level in the soil from 10.15 mg/kg to 6.89 mg/kg in the absence of an SMF, corresponding to a reduction rate of 32.11% ($p < 0.05$). However, in the presence of an SMF, at the end of the experiment, we obtained a mercury concentration of 6.86 mg/kg, which corresponds to a reduction rate of 32.41% ($p < 0.05$). These results show that the non-bioaugmented soil samples exposed to an SMF and those not exposed had almost the same mercury concentrations after 30 days of the experiment ($p > 0.05$). According to these results, we can conclude that the SMF does not affect the mercury reduction rate in the soil not bioaugmented by *P. stutzeri* LBR.

The results of monitoring the rate of mercury reduction in sterile soil not bioaugmented by *P. stutzeri* LBR in the presence and absence of an SMF are presented in Fig 2. After 30 days of incubation at 30°C, the Hg concentration in sterile soil decreased from 7.56 mg/kg to 6.99 mg/kg in samples without an SMF, representing a 7.53% reduction, and to 6.45 mg/kg in samples with SMF, representing a 14.68% reduction ($p < 0.05$). These results indicate that in the absence of *P. stutzeri* LBR bioaugmentation, SMF had a limited effect on mercury bioremediation.

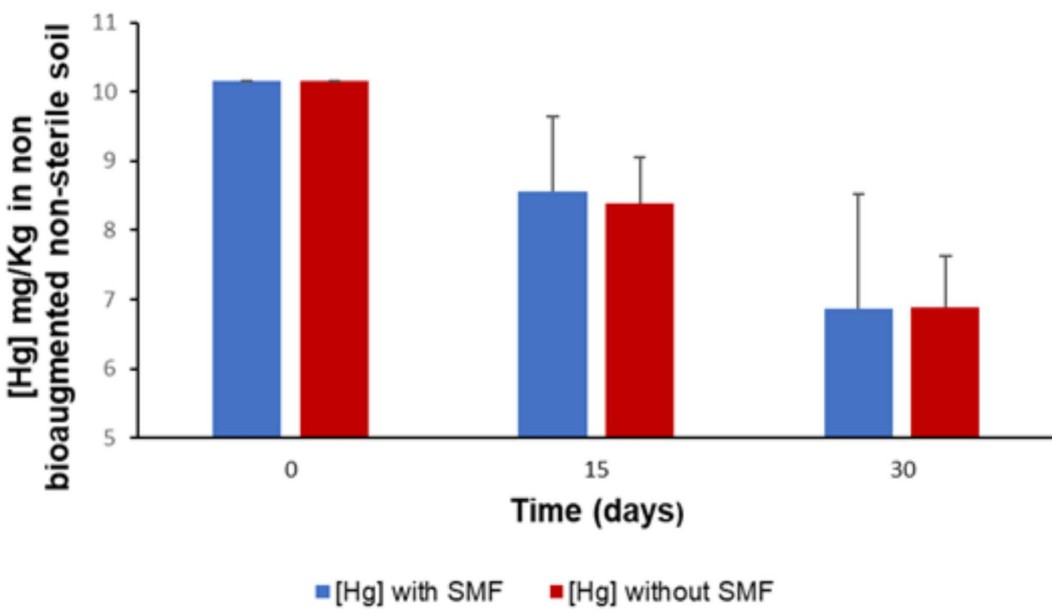

**Fig 1. Kinetics of Hg reduction in non-sterile soil without bioaugmentation.**

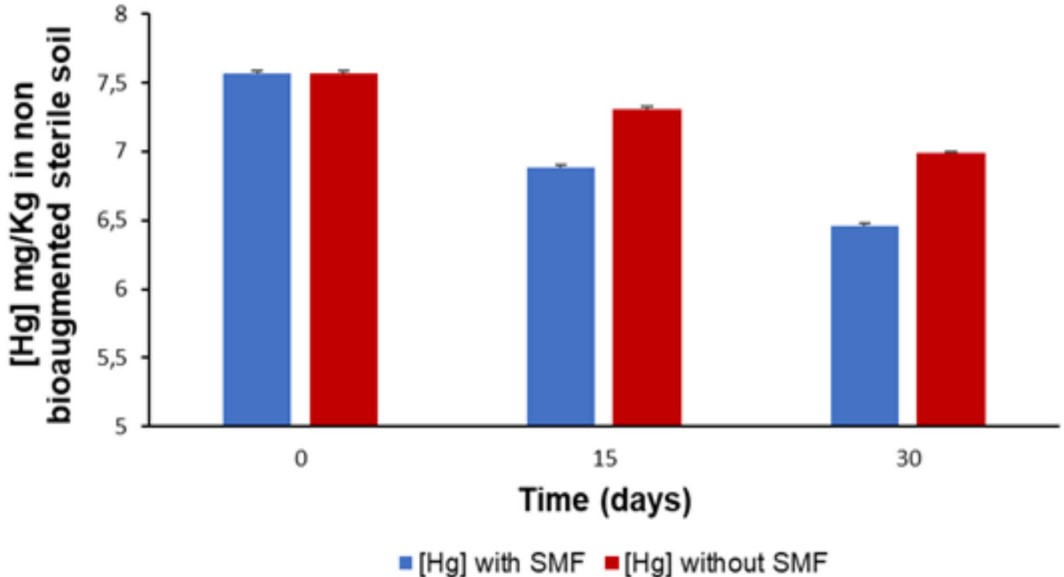

**Fig 2. Kinetics of Hg reduction in non-bioaugmented sterile soil.**

The results of monitoring the rate of mercury reduction in non-sterile soil bioaugmented by *P. stutzeri* LBR in the presence and absence of an SMF are presented in Fig 3.

The results showed a decrease in mercury concentration in the non-sterile soil in the presence of an SMF, dropping from 10.15 mg/kg to 1.85 mg/kg over 30 days, with a reduction rate of 81.77% ($p < 0.05$). During the first 15 days, the total microbial population in the bioaugmented soil gradually increased from $27 \times 10^9$ CFU/mL to $44.66 \times 10^9$ CFU/mL. This

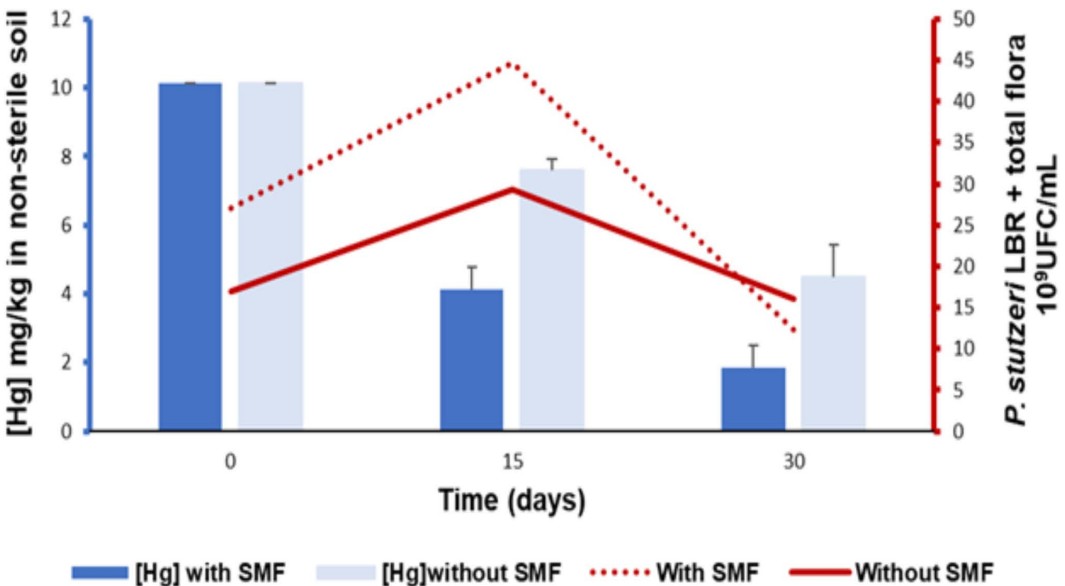

**Fig 3. Monitoring bacterial growth in non-sterile soils bioaugmented with *P. stutzeri* LBR in relation to mercury bioremediation, in the presence and absence of induction by a static magnetic field (SMF).**

increase was followed by a decrease in cell numbers, reaching $12.33 \times 10^9$ CFU/mL. In contrast, the results in the absence of an SMF showed a less pronounced decrease in mercury concentration, from 10.15 mg/kg to 4.47 mg/kg, with a reduction rate of 55.96% ($p < 0.05$). The total soil flora showed an initial growth from $16.93 \times 10^9$ CFU/mL to $29.4 \times 10^9$ CFU/mL during the first 15 days, followed by a decline to $16.03 \times 10^9$ CFU/mL in the absence of an SMF. Based on these results, we conclude that the increase in bacterial biomass was significantly greater in the presence of an SMF, leading to a higher mercury reduction rate ($p < 0.05$). The presence of the *P. stutzeri* LBR strain significantly enhanced mercury reduction ($p < 0.05$). The decline in bacterial numbers at the end of the experiment can be attributed to the depletion of the pyruvate carbon source added to the medium.

The results of monitoring the rate of mercury reduction in sterile soil bioaugmented by *P. stutzeri* LBR in the presence and absence of an SMF are presented in Fig 4.

In the presence of an SMF, Hg concentration dropped from 7.56 mg/kg to 0.97 mg/kg, with a reduction rate of 87.17% ($p < 0.05$). This substantial reduction was accompanied by an increase in the biomass of *P. stutzeri* LBR during the first 15 days of monitoring, rising from $13.66 \times 10^9$ CFU/mL to $17.66 \times 10^9$ CFU/mL, before returning to its initial value ($13.66 \times 10^9$ CFU/mL) by the end of the experiment. In contrast, the samples not exposed to an SMF showed a concentration of 4.11 mg/kg on day 30, with a reduction rate of 45.63%. Specifically, during the first 15 days, the bacterial biomass increased from $18.33 \times 10^9$ CFU/mL at the start of the experiment to $25 \times 10^9$ CFU/mL. However, during the second half of the experiment, there was a notable decrease in bacterial numbers, dropping to $15.66 \times 10^9$ CFU/mL. From these results, it can be concluded that the presence of the *P. stutzeri* LBR strain significantly enhanced mercury reduction ($p < 0.05$).

## Discussion

The use of bioremediation and biotechnological approaches to remediate environmental pollutants, such as mercury, has been proposed as an economically viable and eco-friendly alternative. Consequently, the development of new technological innovations to facilitate effective environmental remediation is a current area of scientific focus.

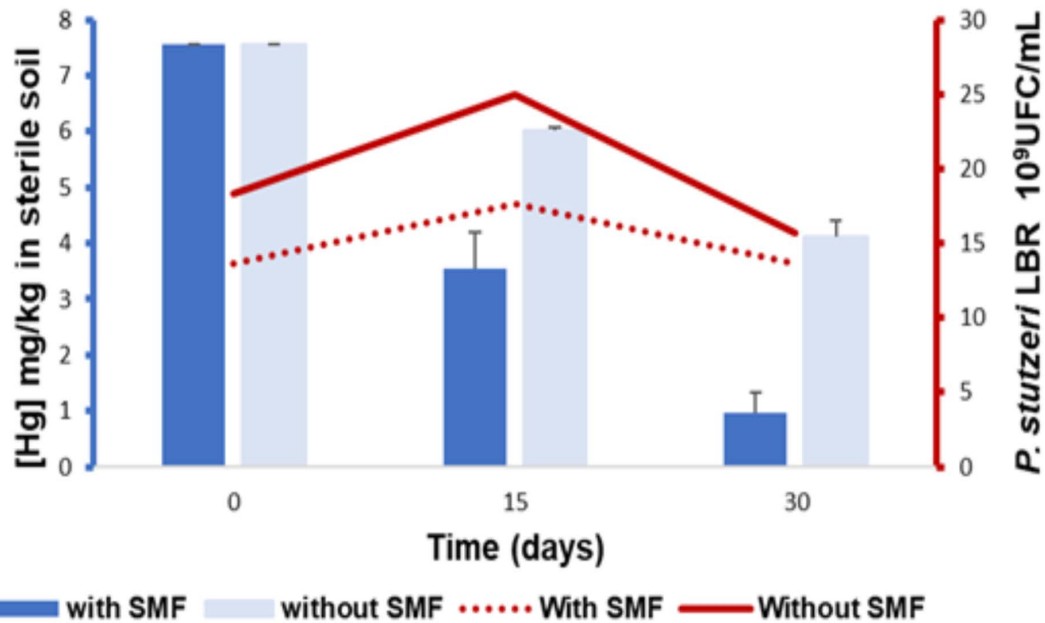

**Fig 4. Monitoring bacterial growth in sterile soils bioaugmented with *P. stutzeri* LBR in relation to mercury bioremediation, in the presence and absence of induction by a static magnetic field (SMF).**

Our research focuses on deploying an SMF in environmental engineering to enhance soil mercury remediation. We aimed to test the combined effect of applying an SMF along with bioaugmentation through the addition of the bacterium *P. stutzeri* LBR on mercury bioremediation. The results demonstrated that, in non-sterile soil, microbial bioaugmentation enhanced mercury bioremediation by 23.85% compared to non-bioaugmented soil ($p < 0.05$). In contrast, in the absence of bioaugmentation, the application of an SMF alone did not produce a significant effect. However, the combined application of an SMF and bioaugmentation increased the bioremediation efficiency by 25.81% compared to bioaugmented soil not exposed to an SMF ($p < 0.05$). Similarly, in sterile soil, microbial bioaugmentation enhanced the bioremediation of mercury by 38.1% compared to non-bioaugmented soil ($p < 0.05$). The application of an SMF alone, in the absence of bioaugmentation, led to a slight increase in bioremediation, with an improvement of 7.15% ($p < 0.05$). In contrast, the combined application of an SMF and bioaugmentation resulted in a 41.54% increase in bioremediation compared to bioaugmented soil not exposed to an SMF ($p < 0.05$). These results demonstrate that microbial bioaugmentation is an effective strategy for enhancing the bioremediation of mercury in both sterile and non-sterile soils. The application of an SMF alone yields limited or negligible effects in the absence of bioaugmentation. However, the combination of an SMF with bioaugmentation leads to a synergistic effect, resulting in a significant improvement in the bioremediation rate. The results showed that, in non-sterile soils, the combination of an SMF, total soil flora, and *P. stutzeri* LBR increased mercury remediation efficiency by 49.36%, compared to only 23.85% in the absence of a synergistic effect between SMF application and soil bioaugmentation. Similarly, in sterile soils, the combination of an SMF and *P. stutzeri* LBR increased mercury remediation efficiency by 72.49%, compared to 38.1% in the absence of a synergistic effect between SMF application and soil bioaugmentation (Fig 5). Thus, the highest mercury remediation rate was obtained in the case of a sterile soil bioaugmented with *P. stutzeri* LBR and exposed to the action of an SMF at the beginning of the experiment.

Sterilized soils demonstrated superior bioremediation efficiency compared to non-sterile soils, which can be attributed to several key factors. Firstly, the absence of microbial competition in sterile soils allows introduced bacteria to proliferate without competition for essential nutrients, thereby optimizing their mercury detoxification activity [44]. In contrast,

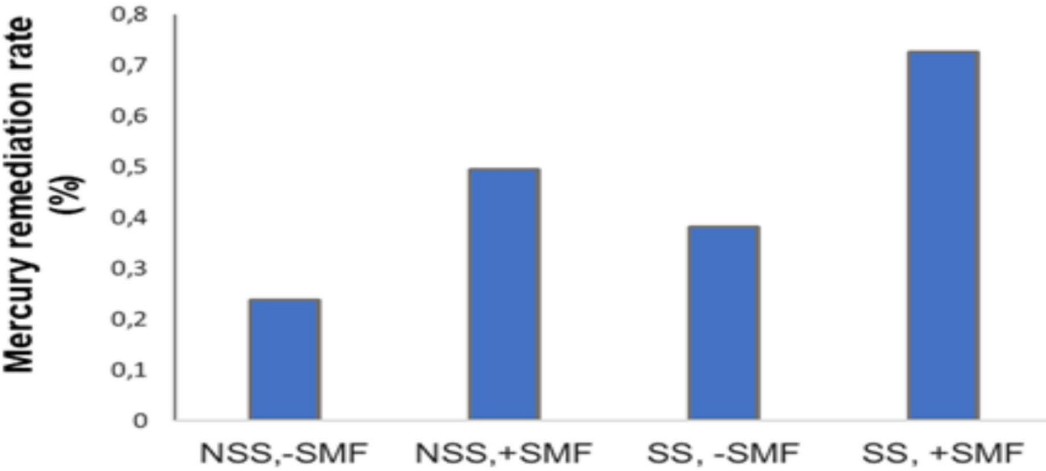

**Fig 5. The impact of a static magnetic field (260 mT) in combination with bioaugmentation on the mercury remediation percentage.** NSS, -SMF = non-sterile soil without SMF; NSS, + SMF = non-sterile soil with SMF; SS, -SMF = sterile soil without SMF; SS, + SMF = sterile soil with SMF.

in non-sterile soils, indigenous microorganisms compete with bioremediating bacteria for vital elements such as carbon, iron, and sulfur, which reduces their detoxification efficiency [45]. Additionally, some native bacteria influence mercury speciation by transforming it into mercury sulfide (HgS), a form with low bioavailability for bacterial detoxification [46]. Furthermore, the production of inhibitory metabolites by certain native strains can hinder the establishment of exogenous mercury-detoxifying bacteria through competitive repression of *mer* promoter activity [47]. Moreover, the presence of protozoan predation in non-sterile soils exerts selective pressure on introduced bacteria, further limiting their survival and activity [48]. Finally, sterilization enables precise control of environmental parameters such as pH, redox potential, and nutrient availability, eliminating biological interferences and creating optimal conditions for the expression of detoxification genes [49].

Furthermore, the results demonstrated a notable increase in bacterial biomass, particularly in the presence of an SMF-induced induction, accompanied by enhanced mercury remediation efficiency. The observed decline in bacterial cells at the conclusion of the experiment can be attributed to the reduction in the carbon source, pyruvate, which was introduced into the medium; however, further analysis, including direct measurements of pyruvate levels, would be necessary to confirm this hypothesis.

In the presence of elevated concentrations of iron (6,504 mg/kg) and chromium (341.03 mg/kg) in the soil, *Pseudomonas stutzeri* LBR maintained efficient Hg bioremediation, likely supported by its intrinsic multimetal resistance mechanisms [50,51]. The simultaneous presence of multiple background metals may promote bacterial adaptive responses, aligning the experimental setting more closely with the environmental complexity of real-world contaminated soils [52].

The study of numerous mercury-contaminated sites has highlighted microorganisms' ability to cope with this type of pollution [7,18]. Mercury bioremediation by bacteria mainly relies on the activity of *mer* genes, which result from the development of tolerance mechanisms through evolution.

Several mercury bioremediation mechanisms in bacteria have been described. The first is biosorption, which corresponds to the accumulation of mercury on the bacterial surface through the secretion of exopolysaccharides (EPS) or biofilm formation. The second mechanism is biovolatilization, which involves the reduction of mercury ($Hg^{2+}$) into a volatile form ($Hg^{o}$). Finally, bioaccumulation corresponds to the sequestration of mercury in intracellular compartments, notably through the production of chelating agents such as metallothioneins [45,53]. As an example, the bacterial strain *Sphingobium* SA2 can eliminate 60% of the mercury present in contaminated soils by combining biovolatilization and

bioaccumulation mechanisms [54]. Similarly, Zheng et al. (2017) showed that the marine strain *P. stutzeri* 273 can resist concentrations of up to 50 µM of mercury and eliminate up to 94% of mercury in culture. Nonetheless, this result was obtained in an artificial LB medium under laboratory conditions, which may not fully reflect the complexity of natural environments. The authors identified several essential genes for this resistance, including *merA*, which encodes a mercuric reductase that converts $Hg^{2+}$ into volatile $Hg^0$, as well as *merT* and *merP*, which encode mercury transport proteins, and *merD*, which encodes a regulatory protein of the *mer* operon [27]. Furthermore, they demonstrated that the presence of mercury in the environment induces bacterial stress, reflected by inhibited flagellar development and biofilm formation. It was also shown that *P. stutzeri* 273 produces a marine exopolysaccharide named EPS273 [27,55]. These results suggest that *P. stutzeri* is capable of accumulating mercury in the form of $Hg^{2+}$ and converting it into $Hg^0$ for volatilization. A similar mechanism is likely involved in the strain *P. stutzeri* LBR, but at this stage, we cannot draw definitive conclusions about the mechanism used.

Another approach involves the use of genetically modified bacteria. This involves introducing a plasmid containing a recombinant gene, which enables the expression of proteins involved in mercury resistance. For instance, the transgenic strain *Cupriavidus metallidurans* MSR33 allows complete volatilization of $Hg^{2+}$ at 0.15 M in a multi-contaminated medium [56]. Similarly, the *Bacillus cereus* BW-03 strain modified with the *mer* operon enables simultaneous biovolatilization and precipitation of mercury, with 100% elimination efficiency in solution [57]. However, although these genetically modified organisms exhibit high performance, they raise ethical and environmental concerns. Their use may also have adverse effects on ecosystems and their integrity [58,59]. It may therefore be preferable to combine bioremediation with a composite microbial system and an SMF to improve remediation efficiency.

The SMF represents an innovative and promising technique for enhancing pollution remediation processes. Initially, its application focused mainly on the bioremediation of wastewater using activated sludge systems. Later, only a few studies investigated its potential in the bioremediation of organic pollutants and heavy metals [30]. Several studies, such as those by Niu et al. (2014) and Zaidi et al. (2014), have shown that SMFs can stimulate microbial activity, leading to increased chemical oxygen demand removal and improved solid-liquid separation [60,61]. The intensity of the magnetic field plays a crucial role: at moderate levels between 7 and 40 mT, SMFs accelerate sludge acclimation, enhance nitrification, and promote biomass growth [31,62,63]. Conversely, higher intensities (360 mT) can inhibit microbial growth, as reported by Zhao et al. (2016) [64]. However, Yavuz and Çelebi (2000) observed microbial growth inhibition starting at 17.8 mT [65]. These findings support the notion that SMFs can stimulate the growth of certain microorganisms while inhibiting that of others. In soils, SMFs with intensities between 0.15 and 0.35 T have been shown to stimulate microbial respiration, thereby promoting the degradation of organic matter [66]. Regarding organic pollutants, the biodegradation of BaP by *Microbacterium maritypicum* CB7 was found to double under a 200 mT SMF [41], while a field strength of 30–60 mT enhanced the degradation of trichloroethylene by 2.4%, enriching bacterial genera such as *Acinetobacter* and *Acidovorax* [67]. As for the bioremediation of heavy metals, the literature reports two noteworthy studies focused on chromium remediation. The application of a weak SMF (optimal at 6.0 mT) enhances Cr(VI) removal efficiency in an anaerobic sequencing batch reactor (ASBR). This enhancement is evidenced by an increase in biomass ranging from 32% to 65% and a reduction of 1–3 hours in the time required to achieve discharge standards [66]. Similarly, a 7 mT SMF promotes Cr(VI) desorption and stimulates the growth of the fungal stain *Geotrichum* sp. in contaminated soils [68]. In summary, these results highlight the potential of an SMF as a valuable ally in microbial bioremediation. However, the efficiency of the process strongly depends on the applied field intensity, the microbial stain, and the type of pollutant, underscoring the need for precise control to achieve targeted, effective, and sustainable decontamination.

Additional experiments will be necessary to deepen our understanding of the biophysical and microbiological interactions that promote mercury bioremediation under the influence of an SMF. In future experiments, it would be worthwhile to conduct molecular and biochemical analyses, such as transcriptomics and proteomics, to investigate the impact of an SMF on gene expression and membrane transport proteins.

We acknowledge that our study represents a preliminary approach at the laboratory scale. Several future research avenues could enhance the practical and ecological relevance of the proposed method, such as optimizing parameters for conducting in situ pilot tests on soils that are genuinely contaminated with mercury, as well as evaluating the impact of SMF application on environmental and ecological factors in the soil.

## Conclusion

The present study aimed to examine the impact of SMFs on the rate of mercury remediation from soil in combination with bioaugmentation. The results showed that, in non-sterile and sterile soils, the combination of an SMF and *P. stutzeri* LBR increased mercury remediation efficiency compared to the absence of a synergistic effect.

In the future, the use of SMFs in combination with bioaugmentation may hold great potential to increase the remediation rate of complex industrial waste and polluted sites. Further research should optimize SMF parameters (intensity, duration, and frequency of application) to evaluate its effectiveness at different scales, both in bioreactors and in real environments.

## Supporting information

**S1 Fig. Picture of the static magnetic field.**
(TIF)

**S1 Table. Detection end quantification Limits of the target Elements (ICP-OES).**
(PDF)

## Acknowledgments

The authors contributed equally to this work. The authors are grateful to Mr Hmida Naoueli, Director of the Laboratory of International Center for Environmental Technologies, Boulevard Leader Yasser Arafat, Tunis, Tunisia. The authors are also thankful to Hayet Hcini, technician at the inorganic chemistry laboratory of the International Center for Environmental Technologies.

## Author contributions

**Conceptualization:** Naima Werfelli, Mariem Taboubi, Sirine Ridene, Ahlem Mansouri, chiraz Abbes.

**Data curation:** Mariem Taboubi.

**Formal analysis:** Naima Werfelli.

**Methodology:** Naima Werfelli, Ahmed Landoulsi, chiraz Abbes.

**Project administration:** chiraz Abbes.

**Resources:** Sirine Ridene, Ahlem Mansouri, Ahmed Landoulsi.

**Software:** Mariem Taboubi, Sirine Ridene, Hadir Bousselmi.

**Supervision:** Hadir Bousselmi, Ahlem Mansouri.

**Validation:** Naima Werfelli, Sirine Ridene, Ahlem Mansouri, Ahmed Landoulsi, chiraz Abbes.

**Visualization:** Hadir Bousselmi, Ahlem Mansouri, Ahmed Landoulsi.

**Writing – original draft:** Naima Werfelli, Mariem Taboubi, Sirine Ridene, chiraz Abbes.

**Writing – review & editing:** Naima Werfelli, Sirine Ridene, Hadir Bousselmi, chiraz Abbes.

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
