## [Decision Letter · Decision Letter 0]

27 Jan 2025

PONE-D-24-56633Novel method for combining microbial bioremediation with static magnetic fields to remediate mercury-contaminated SoilsPLOS ONE

Dear Dr. Abbes,

Thank you for submitting your manuscript to PLOS ONE. After careful consideration, we feel that it has merit but does not fully meet PLOS ONE’s publication criteria as it currently stands. Therefore, we invite you to submit a revised version of the manuscript that addresses the points raised during the review process.

We look forward to receiving your revised manuscript.

Kind regards,

Naga Raju Maddela, Ph.D

Academic Editor

PLOS ONE

Reviewers' comments:

Reviewer's Responses to Questions

**Comments to the Author**

1. Is the manuscript technically sound, and do the data support the conclusions?

Reviewer #1: Partly

Reviewer #2: Yes

2. Has the statistical analysis been performed appropriately and rigorously? 

Reviewer #1: I Don't Know

Reviewer #2: Yes

3. Have the authors made all data underlying the findings in their manuscript fully available?

Reviewer #1: No

Reviewer #2: Yes

4. Is the manuscript presented in an intelligible fashion and written in standard English?

Reviewer #1: No

Reviewer #2: Yes

5. Review Comments to the Author

Reviewer #1: Specific comments to the authors

The writing could benefit from clearer organization. Some sections, e.g. materials and methods, lack clarity, descriptions, and smooth transitions, making it difficult for readers to follow the work done. The study should provide more details on the experimental design, including controls and replicates. This information is crucial for assessing the reliability and validity of the results. The methods of statistical analysis used to evaluate the data should be clearly described. Without this, it is challenging to assess the robustness of the conclusions drawn from the results (specific tests, data transformation, etc.). There should be a stronger connection between the Materials and methods and the Results sections. For instance, specific details about how the treatments were applied should directly correlate with the results presented. Despite the promising findings presented, the work needs to be deeply revised, addressing these suggestions could enhance the clarity, rigor, and impact of the research.

Abstract:

1. I suggest revising the spelling and formatting, respecting the spacing throughout the text.

2. Do not use abbreviations, particularly in the case of bacterial isolate nomenclature.

3. When starting to refer to the results obtained, emphasize the observed effect (increase, decrease, etc.) on Hg removal instead of ‘The results demonstrated the impact of SMF on...’.

4. The statement that the combination of SMF and P. stutzeri ‘accelerated’ metal remediation in sterile soil, compared to SMF alone, suggests that the effect was observed in a shorter treatment time. If this was the case, I suggest clarification, otherwise ‘accelerated’ should be replaced by increased/promoted removal by x %, relative to SMF alone.

Introduction:

I suggest a clear delimitation of the impact of the problematic addressed, an expansion of previous studies (e.g. on the presence of resistance genes in the bacterial isolate used, use of SMF and their main results). I suggest reorganizing the section, including a bibliographical update and finally detailing clearly how the present article aims to contribute to solving the problem in question (mentioned vaguely in lines 75-76, 81-82 and 93-94). Additionally, data should be provided on the Hg levels (ppm, mg/kg) permitted/suggested by any regulatory or health control organism.

In particular:

1- Line 42, where it is argued that mercury is a bioaccumulative ‘toxin’, I suggest revision of the term. Hg can be classified as a ‘toxic’ metal, but not a toxin, which presupposes a biological origin of the compound in question. The above-mentioned citation of reference [1] does not claim that it is a toxin. I suggest adding its classification according to regulatory or health agencies/organizations, e.g. IARC.

2- I suggest broadening the discrimination between the geogenic origin of Hg and its use and consequent redistribution due to anthropogenic activity, perhaps by replacing citation 2 (line 44) or adding new citations. It would also enrich the introduction if the regional and global impact of the problem addressed were highlighted.

3- I suggest reorganizing what is included between lines 48 and 51, replacing citation 6 with a more current one, clarifying the impact of the different states that the metal can have, the predominant forms in the environments of interest and the relative toxicity of each form.

4- Line 59, replace ‘toxicant’ by ‘toxic metal’ or ‘pollutant’.

5- Line 61 mentions the ability of bacteria to convert Hg into ‘non-toxic’ forms, clarify which would be the starting (toxic) forms and those derived from microbiological transformation. Perhaps this should be placed on line 66, after quote 18, where these forms are mentioned without the clarification of their toxicity.

6- Line 63, reference is made to the mer operon as one of the ‘most intriguing’ aspects of resistance to this metal, however its components, mode of action and relationship of the bacterial isolate selected (P. stutzeri) with this operon (if there are gene components previously found/described in this isolate, etc.) are not clearly detailed.

7- Lines 72- 74, P. stutzeri is mentioned as one of the bacterial isolates previously used in Hg remediation studies, I suggest including the main results to be highlighted in these studies, allowing a priori to know the effectiveness of this isolate in the remediation of sites contaminated with this metal. In this point, quotes 25 and 26 should be included (adding concrete removal data), reorganize.

8- The objective of the work should be reformulated (unifying what is stated in lines 75-76, 81-82 and 93-94), detailing the context of Hg bioremediation and the novelty of the application of SMF. To state clearly how the present study intends to contribute to solve the problem of interest.

9- Lines 82-88 Rewrite reference to previous studies with SMF, highlighting the main results (numerical data, % removal, etc.). Line 85 mentions that SMF ‘can accelerate the treatment of water’, rewrite, mentioning removal/remediation and data. A study analyzing the role of SMF in wastewater is mentioned (citation 27), and previously no wastewater was detailed as a source of Hg contamination.

At this point in the introduction, it is not clear which Hg contaminated matrix is of interest to the present work.

10- Lines 89-91 The fact that there are previous studies combining bacterial isolation (P. stutzeri) with SMF for the treatment of contaminated sources (citation 25) detracts from the originality of the work, despite the differential nature - organic vs. inorganic - of the pollutant in question.

11- Line 92 I suggest changing the term ‘decontamination’ to ‘remediation’.

Materials and methods

1- Lines 97-101, give more details about the area sampled, why it was selected, land use history, coordinates, number of samples taken, extent of area sampled.

2- Lines 102-104, detail which properties (physical and chemical) were determined and provide data of the correspondent equipment used.

3- Line 106, detail which heavy metals and major elements are referred to, range of detection/quantification, units of expression of the results.

4- Line 111, clarify whether the analyzer used (DMA-80) measured total mercury or allows the speciation of this to be quantified, brand of equipment.

5- Lines 114-118, up to this point it is not clear why they decided to add Hg instead of working with the reference concentrations in the sampled area, or if they have not quantified the metal in these samples. It is not clear at what concentrations of the metal they want to work with (ppm, M, Mm), by adding 148 mg HgSO4 to 100 ml H2Od you have prepared a 5 mM solution of the metal (or of the salt, whose molecular weight is predominantly contributed by mercury), you should clarify what are the desired working concentrations, why they have been selected and how they were prepared. In line 117, it is not understood what is meant by ‘at 50-gram dose’, as it cannot be equivalent to the concentration of Hg present in 1 kg of soil when adding that solution (as understood in the text). I suggest revising and rewrite providing all the necessary data for the understanding of the selected methodology. I suggest modifying the title of this subsection, I do not consider the term ‘ground doping’ to be appropriate.

6- Lines 119-125, If the bacterial isolation was obtained by working group members, which I deduce from the reference provided, this should be clarified. Which sequences are available in GenBank, 16S?, complete genome sequence?, provide the corresponding accession/bioproject numbers. If there is a history of the presence of genes involved in Hg tolerance/resistance in this isolate add it beforehand in the Introduction section.

7- Lines 127-141, in the bioaugmentation assays the authors should explain why they do not use the liquid inoculum of the bacteria (or sterile medium in the corresponding treatments) and do the washes mentioned in lines 127 and 128. In this respect, the composition of the media used (g/L), incubation/growth conditions of the bacteria, CFU/ml of the culture that proceeded to centrifugation should be included. This centrifugation was in duplicate? what is the volume of growth media centrifuged?, what was the total bacterial count used?, 7x1010 CFU/ml, 1.75x109 CFU/ml?. In line 130 add the abbreviation ‘rpm’.

8- Line 132: 20 mg of soil in 40 ml of medium (with or without bacteria) corresponds mainly to a test in liquid medium (the volume of soil used is almost negligible), which considerably biases the conclusions that can be drawn for this matrix, on which the results are presented in the Abstract and throughout the work.

9- From what is written in lines 134 to 136 I interpret that there are 3 different treatments (in flask 1, flask 2, flasks 3+4), in this description the sterile soil without bioaugmentation is not included as a control, which I suggest adding. So far it is interpreted that the n of treatment replicates is 1 (for the first two treatments) and 2 for the non-sterile soil without bioaugmentation (flasks 3 and 4), which makes it very weak statistically. Whether what is stated in line 139 corresponds to replicates of these treatments should be clarified above, or whether it refers to temporal replicates of the trials. It is also not clarified that the soil sample used corresponds to the soil added with Hg, clarify, explain in detail all the procedures so that those who did not carry out the work can clearly understand the methodological steps followed.

10- Lines 138-139 ‘Bacterial counts and mercury analysis were conducted on a weekly basis’, detail here the methodology used for this purpose, did they quantify total Hg, speciation? If these parameters were measured weekly for a total period of 28 days there should be 4 sampling and data collection times (does not correspond with the times shown in Fig. 1-4), please clarify.

11- Lines 140-141, to which aseptic conditions do you refer? in which procedure were they used? through which materials or equipment were they achieved?

12- Lines 142-152, describe the equipment used to expose the samples to MFS, I suggest attaching a picture of it.

13- Lines 152-153 ‘prior to the commencement of the observation period’, refers to the soil samples prior to disposition in the previous bioaugmentation test, it is not easily detectable if it was the same test (as interpreted in the Abstract) or not, needs to be clarified, reformulated, at what time the soil samples were subjected to this field, pre or post dilution in the 40 ml of medium, I suggest to attach a picture of it.

14- Lines 156-162, why are treatments described again, can this section not be summarily unified with the one previously describing the different treatments in the Flasks, why are they now ‘Bottles’ 1-4, when I understand that they refer to the treatments already described.

Results

1- I suggest presenting the different results in accordance with what is described in the Materials and Methods section, without the divisions presented in the subtitles of lines 202-203, 224-225, 246.

2- Line 223 mentions a treatment that was not described in the Materials and Methods section, ‘non-bioaugmented sterile soil’.

Discussion

1- Line 264, I suggest replacing the term ‘decontamination’ with remediation.

2- Do you consider SMF as an ‘economically viable’ methodology? in which cases? for which matrices? would it be possible to use it on a large scale? discuss these, pros and cons of this methodology, potential impact on native microbiota. It remains to add to this section previous work carried out to remediate Hg, its main results, and contrasts with what is observed in the current work. A considerable part of the discussion refers to other contaminants, which are not the one of interest, nor is there any discussion on the mechanisms that allow P. stutzeri LBR to remediate the metal.

Reviewer #2: I have read the present manuscript authored by Naima Werfelli and colleagues with great interest and attention to detail. This manuscript explores a novel approach to enhancing microbial bioremediation by incorporating static magnetic fields to address mercury contamination in soils. Mercury, a persistent environmental pollutant, poses significant risks to ecosystems and human health.

Attached, I have provided some comments and suggestions that could help improve the article.

1. Consider reorganizing the content to progress from general mercury contamination issues to the specific focus of your study. For example:

• Mercury as an environmental pollutant (current impacts and sources).

• Current remediation challenges (limitations of traditional methods).

• Potential of bioremediation (your approach and objectives).

2. Some sentences are lengthy and contain multiple ideas. Consider breaking them down to improve clarity. For example: You mention that "a variety of bacteria have the capacity to resist toxic forms of mercury and subsequently convert them into non-toxic forms."

• It would be helpful to briefly explain the mechanisms involved in these conversions.

• You could also mention if P. stutzeri LBR has been compared with other strains for effectiveness.

3. The study objective is stated clearly at the end, but consider emphasizing what practical applications or next steps could follow from this research.

4. The material and method section is well-structured, but it could benefit from clearer subheadings and a more logical flow. Consider structuring it as follows for better readability:

• Soil Sampling and Preparation

• Soil Analysis (Physical, Chemical, and Heavy Metal Concentration)

• Bacterial Strain and Inoculation

• Bioaugmentation Tests

• Static Magnetic Field (SMF) Application

• Monitoring and Analysis

• Statistical Analysis

5. Specify if the mercury solution was mixed uniformly to ensure even distribution in the soil.

• Clarify why pH 7 was chosen and its significance in mercury solubility/stability.

• Indicate whether control groups were included for comparative purposes.

• The bacterial inoculum concentration (70 x 10⁹ cells) should mention if it was determined through OD measurements or CFU counting for better reproducibility.

• Mention if post-hoc tests were applied to detect significant differences between groups.

• Instead of "significance level of p < 0.05," consider rephrasing to "statistical significance was set at p < 0.05."

6. Ensure correct citation formatting, e.g.,

• "[4 , 5]" should be "[4,5]" (remove spaces before commas).

• Proper reference order for standards like "AFNORNF ISO 11265" should follow a uniform style.

• Some paragraphs contain excessive detail; simplifying them can make the section more reader-friendly.

• • Increase the clarity of tables and charts.

7. Summarize the key findings more clearly, highlighting the effectiveness of bioaugmentation and SMF's impact.

8. When presenting comparative results, such as the 49.36% and 72.49% remediation rates, consider briefly explaining why sterile soils showed a higher improvement compared to non-sterile soils. This helps provide context for the reader.

9. Clarify mechanisms where necessary; for instance, when discussing microbial competition in non-sterile soils, elaborate briefly on how this impacts mercury uptake.

10. Instead of saying "published studies," specify which studies to strengthen credibility.

11. A brief mention of potential future experiments to confirm these mechanisms could enhance the scientific rigor of the document.

12. Expand on future research opportunities, such as optimizing SMF parameters (intensity, duration), exploring its application in real-world contaminated sites, and assessing long-term environmental impacts.

13. Ensure uniform formatting in terms of punctuation, spaces, and abbreviations.

14. For example, journal names should follow a consistent format (e.g., Atmos. Environ. or Atmospheric Environment, but not a mix of both).

• Instead of writing the DOI as a URL, it's better to present it in the standard format: doi:10.xxxx/xxxx.

• Example:

https://doi.org/10.1016/S1352-2310(97)00293-8 → doi:10.1016/S1352-2310(97)00293-8

• Ensure a consistent style for author initials (e.g., "W.H. Schroeder" vs. "WH Schroeder").

• Remove unnecessary spaces between initials.

• Ensure all references have complete citation details, such as volume, issue, page numbers, and publication year.

6. PLOS authors have the option to publish the peer review history of their article (what does this mean? ). If published, this will include your full peer review and any attached files.

**Do you want your identity to be public for this peer review?** For information about this choice, including consent withdrawal, please see our Privacy Policy .

Reviewer #1: No

Reviewer #2: No

---

## [Author Response · Author response to Decision Letter 1]

16 Jun 2025

Dear Editor-in-Chief of PLOS ONE,

Thank you very much for your interest,

Authors are also thankful for reviewers for their interest and precious comments,

Please find below the answers to reviewer comments:

Note:

Line order was changed between the old manuscript and the revised manuscript.

In response to the reviewers’ comments, we have revised the manuscript accordingly.

The changes made following Reviewer 1’s suggestions are highlighted in yellow, and those addressing Reviewer 2’s feedback are highlighted in blue.

Review Comments to the Authors

Reviewer #1:

Abstract

1. I suggest revising the spelling and formatting, respecting the spacing throughout the text.

We thank reviewer 1 for this comment, the manuscript was re-read by the Proof-Reading Service (please view the certificate).

2. Do not use abbreviations, particularly in the case of bacterial isolate nomenclature.

Thank you for your comment. I have removed all abbreviations from the abstract, particularly those related to bacterial isolate nomenclature, to improve clarity.

3. When starting to refer to the results obtained, emphasize the observed effect (increase, decrease, etc.) on Hg removal instead of ‘The results demonstrated the impact of SMF on...’.

We thank the reviewer 1 for this comment; the sentence " The results demonstrated the impact of SMF on..." is replaced by the sentence " Mercury remediation was enhanced by SMF ……." at line 32 of the abstract.

4. The statement that the combination of SMF and P. stutzeri ‘accelerated’ metal remediation in sterile soil, compared to SMF alone, suggests that the effect was observed in a shorter treatment time. If this was the case, I suggest clarification, otherwise ‘accelerated’ should be replaced by increased/promoted removal by x %, relative to SMF alone.

We thank Reviewer 1 for this comment. In the revised manuscript, we have replaced the word ‘accelerated’ with ‘increased’ (as suggested) on line 34 .

Introduction :

1- Line 42, where it is argued that mercury is a bioaccumulative ‘toxin’, I suggest revision of the term. Hg can be classified as a ‘toxic’ metal, but not a toxin, which presupposes a biological origin of the compound in question. The above-mentioned citation of reference [1] does not claim that it is a toxin. I suggest adding its classification according to regulatory or health agencies/organizations, e.g. IARC.

We thank Reviewer 1 for this comment. In the revised manuscript, we have replaced the word ‘toxin’ with ‘toxic’ (as suggested) on line 45. We have included the classification of mercury in accordance with the guidelines provided by IARC and UNEP at line 49.

2- I suggest broadening the discrimination between the geogenic origin of Hg and its use and consequent redistribution due to anthropogenic activity, perhaps by replacing citation 2 (line 44) or adding new citations. It would also enrich the introduction if the regional and global impact of the problem addressed were highlighted.

Thank you for your valuable feedback. I would like to confirm that I have carefully addressed your comment regarding the sources and impacts of mercury. As suggested, I have included references to both natural sources (e.g., volcanic eruptions and weathering of rocks) and anthropogenic activities (e.g., coal combustion and mining) based on the works of Sonke et al. (2023). Additionally, I have emphasized the global dispersion and bioaccumulation of mercury, highlighting its environmental and health risks, as supported by Outridge et al. (2018) and UNEP (2018). The corrections have been made in the manuscript between line 50 and line 53.

3- I suggest reorganizing what is included between lines 48 and 51, replacing citation 6 with a more current one, clarifying the impact of the different states that the metal can have, the predominant forms in the environments of interest and the relative toxicity of each form.

Thank you for your insightful suggestion. I have carefully reorganized the content between lines 48 and 51 as recommended. Additionally, I have replaced citation 6 with more current and relevant Kumar et al. (2024). To address your comment, I have clarified the impact of the different states that mercury can have, specified the predominant forms in the environments of interest, and discussed the relative toxicity of each form. These changes aim to provide a clearer and more up-to-date discussion of the topic. The corrections have been made in the manuscript between line 54 and line 69.

4- Line 59, replace ‘toxicant’ by ‘toxic metal’ or ‘pollutant’.

We thank Reviewer for this comment. Following the revisions in the text, this word has been removed.

5- Line 61 mentions the ability of bacteria to convert Hg into ‘non-toxic’ forms, clarify which would be the starting (toxic) forms and those derived from microbiological transformation. Perhaps this should be placed on line 66, after quote 18, where these forms are mentioned without the clarification of their toxicity.

Thank you for your valuable comment. we have carefully clarified the toxic forms of mercury (such as methylmercury and inorganic Hg²⁺) and the non-toxic or less toxic forms resulting from microbiological transformation (e.g., elemental mercury Hg⁰). As you recommended, we have also repositioned this clarification to line 66, after citation 18, to ensure a more logical and coherent presentation of the information. The corrections have been made in the manuscript between line 74 and line 85.

6- Line 63, reference is made to the mer operon as one of the ‘most intriguing’ aspects of resistance to this metal, however its components, mode of action and relationship of the bacterial isolate selected (P. stutzeri) with this operon (if there are gene components previously found/described in this isolate, etc.) are not clearly detailed.

We thank the reviewer for this relevant suggestion. We have expanded the section on the mer operon by detailing its components, mode of action, and its connection with Pseudomonas stutzeri. We have added an explanation of the merA, merR, merT, merP, and merB genes, as well as their role in mercury detoxification. Additionally, we have mentioned that P. stutzeri possesses these genes, which grants it a demonstrated ability to resist and detoxify mercury. The corrections have been made in the manuscript between line 91 and line 96.

7- Lines 72- 74, P. stutzeri is mentioned as one of the bacterial isolates previously used in Hg remediation studies, I suggest including the main results to be highlighted in these studies, allowing a priori to know the effectiveness of this isolate in the remediation of sites contaminated with this metal.

Thank you for your comment. In this study, we present the first investigation of mercury degradation by the P. stutzeri LBR strain selected in our laboratory. In contrast, the Zheng et al. (2017) demonstrated that the marine strain P. stutzeri 273 can resist mercury concentrations up to 50 µM and eliminate up to 94% of mercury in culture. However, this degradation rate was measured in an artificial LB medium, not under real environmental conditions such as marine sediments or contaminated seawater. The corrections have been made in the manuscript between line 94 and line 101.

In this point, quotes 25 and 26 should be included (adding concrete removal data), reorganize.

Thank you for your comment. We have included quotes 25 and 26, added the concrete removal data, and reorganized the section as suggested. “Specifically, it was capable of degrading 87% of 1,1,1-trichloro 2,2-bis(p-chlorophe´nyl) ethane (DDT) and 83% of Benzo(a)pyrene (BaP) in the environment [25]. Moreover, when bioaugmented individually in non-sterile soil, it effectively reduced Pb concentrations by 58.07% [26]. Following the modifications made to the manuscript, the line numbers and references have changed. The corrections have been made in the manuscript between line 103 and line 106.

8- The objective of the work should be reformulated (unifying what is stated in lines 75-76, 81-82 and 93-94), detailing the context of Hg bioremediation and the novelty of the application of SMF. To state clearly how the present study intends to contribute to solve the problem of interest.

Thank you for your comment. As recommended, I have reworded the study objective between lines 116 and 124 in the revised manuscript.

9- Lines 82-88 Rewrite reference to previous studies with SMF, highlighting the main results (numerical data, % removal, etc.).

Line 85 mentions that SMF ‘can accelerate the treatment of water’, rewrite, mentioning removal/remediation and data. A study analyzing the role of SMF in wastewater is mentioned (citation 27), and previously no wastewater was detailed as a source of Hg contamination. At this point in the introduction, it is not clear which Hg contaminated matrix is of interest to the present work.

We thank the reviewer for this valuable comment. Since the primary focus of this study is polluted soil, we have clarified this point at the beginning of the paragraph (line 108) before discussing previous studies that applied SMF for the bioremediation of soils and wastewater.

We have changed [can accelerate the treatment of water] to [enhanced wastewater remediation] (Line 112).

We have revised this section to include quantitative results from previous SMF studies (Line 109 to 115)

10- Lines 89-91 The fact that there are previous studies combining bacterial isolation (P. stutzeri) with SMF for the treatment of contaminated sources (citation 25) detracts from the originality of the work, despite the differential nature - organic vs. inorganic - of the pollutant in question.

Citation 25 (corresponds to citation 28 in the revised manuscript) refers to the work of Mansouri et al. (2019), conducted in our laboratory. The difference between that study and the current work is not limited to the nature of the pollutants investigated (organic vs. inorganic), but lies more importantly in our aim to understand how a static magnetic field (SMF) can influence distinct metabolic pathways involved in bioremediation. Indeed, it is well known that bacterial remediation mechanisms differ depending on the type of pollutant: in the case of an organic pollutant, bacteria can use it as a carbon source, whereas for a heavy metal, they tend to accumulate, transform, or precipitate it. Furthermore, bioremediation mechanisms can vary significantly not only from one heavy metal to another, but also between different bacterial species. Moreover, very limited data are currently available on this subject, which we consider to be highly promising. The mechanisms by which SMF influences bioremediation remain poorly understood, and we believe it is essential to further investigate how a static magnetic field can accelerate bioremediation. Which pathways are involved? How is it possible to enhance bioremediation using two different types of pollutants (organic and inorganic), despite the fact that the metabolic mechanisms involved are fundamentally different?

In conclusion, based on all these arguments, we believe that this work is original and fits well within the scope of this project, which our team considers of high interest, as it contributes to a better understanding of the mechanisms of SMF action in bioremediation, with potential applications in the future.

11- Line 92 I suggest changing the term ‘decontamination’ to ‘remediation’.

We thank the reviewer for this relevant suggestion. In the revised manuscript, we have replaced the word ‘decontamination ’ with ‘remediation’ (as suggested) on line 119.

Materials and Methods

1- Lines 97-101, give more details about the area sampled, why it was selected, land use history, coordinates, number of samples taken, extent of area sampled.

Thank you for your valuable comment. We have revised Lines 97-101 to provide more details about the sampled area as requested. The soil samples were collected in the Kasserine region, located in the west-central part of Tunisia, within an industrial zone (GPS: 35.168534, 8.817806). This site was chosen due to its historical industrial activity for the purpose of industrial waste disposal for more than two decades. A total of 30 soil samples were collected over an area of approximately 300 m² to ensure a representative analysis of the site. These additional details have been incorporated into the revised manuscript to improve clarity and context (Line 128 to 132).

2- Lines 102-104, detail which properties (physical and chemical) were determined and provide data of the correspondent equipment used.

Thank you for your comment. The requested details (physical/chemical properties and corresponding equipment specifications) have been added in Lines 148 to 161 of the revised manuscript.

3- Line 106, detail which heavy metals and major elements are referred to, range of detection/quantification, units of expression of the results.

We thank the reviewer for this valuable comment. In response, we have detailed the heavy metals and major elements referred to in the study (Line 162 to 163), including the units used to express the results (mg/kg dry matter) Line 167. Table 1 below presents the detection and quantification limits for the analyzed elements by (ICP-OES) This table will be included in the Supporting Information section of the article. The LOD and LOQ values for mercury measured by the DMA-80 are included at line 170 to 171 of the revised manuscript.

4- Line 111, clarify whether the analyzer used (DMA-80) measured total mercury or allows the speciation of this to be quantified, brand of equipment.*

We thank the reviewer for this comment. We confirm that the DMA-80 analyzer measures total mercury, and the brand of the equipment is Milestone. This clarification has been added to the manuscript for accuracy (Line 168 to 171).

5- Lines 114-118, up to this point it is not clear why they decided to add Hg instead of working with the reference concentrations in the sampled area, or if they have not quantified the metal in these samples. It is not clear at what concentrations of the metal they want to work with (ppm, M, Mm), by adding 148 mg HgSO4 to 100 ml H2Od you have prepared a 5 mM solution of the metal (or of the salt, whose molecular weight is predominantly contributed by mercury), you should clarify what are the desired working concentrations, why they have been selected and how they were prepared. In line 117, it is not understood what is meant by ‘at 50-gram dose’, as it cannot be equivalent to the concentration of Hg present in 1 kg of soil when adding that solution (as understood in the text). I suggest revising and rewrite providing all the necessary data for the understanding of the selected methodology. I suggest modifying the title of this subsection, I do not consider the term ‘ground doping’ to be appropriate.

We thank the reviewer for this comment. After analyzing the mercury content in the soil from the sampling area (soil S1), we detected a very low mercury concentration of 0.3 mg/kg DM (see Table 1 in the revised manuscript). In fact, natural mercury concentrations in uncontaminated soils typically range from 0.003 and 0.5 mg/kg DM Arbestain et al. (2OO9). In residential soils, intervention thresholds, although they vary across national regulations, generally converge around values between 7 and 10 mg/kg DM (Bell 2016 , UNEP 2013). These findings indicate that soil S1 was not contaminated with mercury. Therefore, we artificially added mercury to achieve adequate concentrations for bioremediation. The explanations have been made in the manuscript : (Line 136 to 137) and (Line 249 to 258).

Additionally, as you suggested, we have changed the section title from “Ground doping” to “Soil sampling and preparation” (Line 127).

6- Lines 119-125, If the bacterial isolation was obtained by working group members, which I deduce from the reference provided, this should be clarified. Which sequences are available in GenBank, 16S?, complete genome sequence?, provide the corresponding accession/bioproject numbers. If there is a history of the presence of genes involved in Hg tolerance/resistance in this isolate add it beforehand in the Introduction section.

We thank the reviewer for this important comment. We confirm that the ba

---

## [Decision Letter · Decision Letter 1]

27 Jul 2025

PONE-D-24-56633R1Novel method for combining microbial bioremediation with static magnetic fields to remediate mercury-contaminated SoilsPLOS ONE

Dear Dr. Abbes,

Thank you for submitting your manuscript to PLOS ONE. After careful consideration, we feel that it has merit but does not fully meet PLOS ONE’s publication criteria as it currently stands. Therefore, we invite you to submit a revised version of the manuscript that addresses the points raised during the review process.

We look forward to receiving your revised manuscript.

Kind regards,

Naga Raju Maddela, Ph.D

Academic Editor

PLOS ONE

Journal Requirements:

Reviewers' comments:

Reviewer's Responses to Questions

**Comments to the Author**

1. If the authors have adequately addressed your comments raised in a previous round of review and you feel that this manuscript is now acceptable for publication, you may indicate that here to bypass the “Comments to the Author” section, enter your conflict of interest statement in the “Confidential to Editor” section, and submit your "Accept" recommendation.

Reviewer #3: (No Response)

2. Is the manuscript technically sound, and do the data support the conclusions?

Reviewer #3: Yes

3. Has the statistical analysis been performed appropriately and rigorously? 

Reviewer #3: Yes

4. Have the authors made all data underlying the findings in their manuscript fully available?

Reviewer #3: No

5. Is the manuscript presented in an intelligible fashion and written in standard English?

Reviewer #3: Yes

6. Review Comments to the Author

Reviewer #3: Review Comments to the Author

Overall Assessment

This study presents a novel and promising approach to mercury bioremediation using Pseudomonas stutzeri LBR combined with static magnetic fields (SMF). The methodology is largely rigorous, and the findings could significantly advance sustainable soil decontamination strategies. However, revisions are required to address inconsistencies, clarify methods, and ensure compliance with journal policies.

Major Technical Concerns

Inconsistent Results Reporting (Page 2, Lines 33–39; Page 10–12)

The Abstract states SMF accelerated remediation by 49.36% (non-sterile) and 72.49% (sterile). In contrast, Results sections report 81.77% (non-sterile) and 87.17% (sterile) reductions in Hg concentration.

Action: Reconcile these discrepancies. Define how "acceleration" is calculated (e.g., percentage increase in remediation rate vs. control). Provide a unified metric throughout.

Unclear Experimental Controls (Page 7, Lines 133–141)

Flasks 3–4 are described as "non-sterile soil + M9" without specifying if P. stutzeri LBR was included. This ambiguity complicates interpretation of bioaugmentation effects.

Action: Explicitly label control groups (e.g., *"Flask 3: Non-sterile soil + M9 (no bacteria)"*).

Unsupported Mechanistic Claims (Page 15, Lines 331–334)

The proposal that SMF enhances mercuric reductase (MerA) activity lacks experimental validation (e.g., enzyme assays).

Action: Include MerA activity data under SMF exposure OR reframe this as a testable hypothesis (e.g., "SMF may stimulate MerA, warranting further study").

Methodology & Reporting Improvements

Ambiguous Hg Dosing (Page 6, Line 118)

The protocol (148 mg HgSO₄ in 100 mL solution added to 1 kg soil) does not specify the final soil Hg concentration.

Action: State Hg concentration post-doping (e.g., "resulting in X mg Hg/kg soil").

Background Metal Levels (Page 9, Table 1)

Elevated Fe (6504 mg/kg) and Cr (341.03 mg/kg) suggest potential soil contamination. Their impact on Hg remediation is unaddressed.

Action: Briefly discuss whether background metals influence Hg bioremediation dynamics.

Data Availability (Throughout)

Raw data (bacterial counts, Hg measurements) are not deposited in a public repository.

Action: Add a Data Availability Statement (e.g., "Data are available in [repository name] at [DOI/link]").

GenBank Accession (Page 6, Line 125)

The P. stutzeri LBR sequence is cited as deposited in GenBank, but the accession number is missing.

Action: Provide the GenBank accession number in the Methods.

Language & Presentation

Terminology & Notation:

Italicization: Inconsistent formatting of bacterial names (e.g., Pseudomonas stutzeri vs. Pseudomonas stutzeri).

Action: Italicize all genus/species names (e.g., P. stutzeri LBR).

Exponents: Bacterial counts use "10º" (e.g., 27 × 10º CFU/mL) instead of "10⁹" (Page 14, Lines 298–302).

Action: Correct to "10⁹" throughout.

Redundancy: Phrases like "non-bioaugmented non-sterile soil" (Page 10, Lines 210–214) are repetitive.

Action: Simplify to "non-bioaugmented soil."

Typos:

"borne on plasmids" → "born on plasmids" (Page 3, Line 64).

Ethics & Policies

Funding Declaration (Page 16, Line 345):

Funding sources are not acknowledged.

Action: Declare funding (e.g., "This work was supported by [Agency]") or state "No funding was received."

Recommendations

Strengths: Innovative SMF application, robust bioremediation outcomes (>80% Hg reduction), and clear potential for environmental impact.

Revisions Needed: Address major technical inconsistencies (Point 1), clarify methods (Points 2, 4, 7), and comply with data-sharing policies (Point 6).

Suggested Minor Edits: Improve terminology consistency (Point 8) and correct typos (Point 9).

Decision: Major Revision required prior to reconsideration.

7. PLOS authors have the option to publish the peer review history of their article (what does this mean? ). If published, this will include your full peer review and any attached files.

**Do you want your identity to be public for this peer review?** For information about this choice, including consent withdrawal, please see our Privacy Policy .

Reviewer #3: **Yes: ** Dr. Mutiu A. Alabi

---

## [Author Response · Author response to Decision Letter 2]

4 Aug 2025

Response to reviewer

Dear Editor-in-Chief of PLOS ONE,

Thank you very much for your interest.

The authors would also like to thank the reviewer for their interest and valuable comments.

Below are our responses to Reviewer 3’s comments:

Reviewer 3

Technical quality

1. Inconsistent Results Reporting (Page 2, Lines 33–39; Page 10–12)

The Abstract states SMF accelerated remediation by 49.36% (non-sterile) and 72.49% (sterile). In contrast, Results sections report 81.77% (non-sterile) and 87.17% (sterile) reductions in Hg concentration.

Action: Reconcile these discrepancies. Define how "acceleration" is calculated (e.g., percentage increase in remediation rate vs. control). Provide a unified metric throughout.

Thank you for your comment. To better explain the percentages obtained, you will find below a summary table of the key results.

Mercury reduction rate (%) Percentage increase mercury in bioremediation efficiency (%)

Without SMF With SMF Without SMF With SMF

Non-sterile soil Non- bioaugmented 32.11 32.42 55.96 - 32.11=

23.85

81.77- 32.42=

49.36

bioaugmented 55.96 81.77

Sterile soil Non-bioaugmented 7.53 14.68 45.63 -7.53 =

38.1 87.17- 14.68=

72.49

bioaugmented 45.63 87.17

The SMF enhances the efficiency of bioremediation in both types of soil. In non-sterile soil, in the absence of SMF, bioaugmentation results in only a 23.85% improvement, compared to 49.36% in the presence of SMF. Similarly, in sterile soil, the improvement increases from 38.1% without SMF to 72.49% with SMF. Thus, SMF acts as an amplifier of bioremediation efficiency, particularly in sterile soil. However, in the absence of bioaugmentation (introduced bacteria) or indigenous microbial flora, the effect of SMF remains limited. It can therefore be concluded that the synergy between SMF and bioaugmentation is essential to optimize and accelerate mercury bioremediation.

2. Unclear Experimental Controls (Page 7, Lines 133–141)

Flasks 3–4 are described as "non-sterile soil + M9" without specifying if P. stutzeri LBR was included. This ambiguity complicates interpretation of bioaugmentation effects.

Action: Explicitly label control groups (e.g., *"Flask 3: Non-sterile soil + M9 (no bacteria)"*).

Thank you for your comment. We have added ‘without P. stutzeri LBR’ to the composition of flasks 3 and 4. Please see line 203 and 204.

3. Page 12, Lines 242–245:

Issue: Attribution of bacterial decline to pyruvate depletion is speculative without supporting data (e.g., pyruvate measurements). Recommendation: Provide data on carbon source depletion or temper conclusions.

Thank you for the comment. We have tempered the conclusion accordingly by adding the sentence: "However, further analysis, including direct measurements of pyruvate levels, would be necessary to confirm this hypothesis." Ligne 381.

4. Unsupported Mechanistic Claims (Page 15, Lines 331–334)

The proposal that SMF enhances mercuric reductase (MerA) activity lacks experimental validation (e.g., enzyme assays).

Action: Include MerA activity data under SMF exposure OR reframe this as a testable hypothesis (e.g., "SMF may stimulate MerA, warranting further study").

We fully agree with the reviewer’s comment. However, we could not locate the statement regarding the effect of SMF on MerA activity in the indicated lines (331–334) or elsewhere in the manuscript.

Language et clarity

1. Throughout: o Issue: Inconsistent italicization of bacterial names (e.g., Pseudomonas stutzeri vs. Pseudomonas stutzeri). o Recommendation: Italicize all genus/species names (e.g., P. stutzeri LBR).

Thank you for your comment. The formatting of all genus and species names has been checked and corrected to ensure consistent italicization throughout the manuscript.

2. Page 3, Line 64:

Issue: "borne on plasmids" → should be "born on plasmids."

Correction: Replace with "born."

Thank you for the comment. The expression has been updated to "born on plasmids" as requested, and the correction has been made in line 80 of the revised manuscript.

3. Page 10, Lines 210–214:

Redundant phrasing ("non-bioaugmented non-sterile soil").

Correction: Simplify to "non-bioaugmented soil."

Thank you for the suggestion. The phrase has been revised to "in non-sterile soil without bioaugmentation" in the revised manuscript, Line 274.

4. Page 13, Lines 268–269: o Issue: Awkward syntax ("deployment of SMF in environmental engineering with the objective of..."). o Correction: "We deployed SMF in environmental engineering to enhance soil mercury remediation."

Thank you for your suggestion. We replaced the sentence with:

"Our research focuses on deploying an SMF in environmental engineering to enhance soil mercury remediation." Line 331.

Methodology & Reporting

1. Ambiguous Hg Dosing (Page 6, Line 118)

The protocol (148 mg HgSO₄ in 100 mL solution added to 1 kg soil) does not specify the final soil Hg concentration.

Action: State Hg concentration post-doping (e.g., "resulting in X mg Hg/kg soil").

Thank you for your comment. We have added the final mercury concentration (10.15 mg Hg/kg soil) to the protocol description line 145 as requested.

2. Background Metal Levels (Page 9, Table 1)

Elevated Fe (6504 mg/kg) and Cr (341.03 mg/kg) suggest potential soil contamination. Their impact on Hg remediation is unaddressed.

Action: Briefly discuss whether background metals influence Hg bioremediation dynamics.

Thank you for this very relevant and insightful comment. The use of multi-contaminated matrices provides a relevant methodological advantage, as it more closely simulates the natural conditions of polluted soils, where bacterial communities are commonly exposed to a complex mixture of heavy metal pollutants.

In this study, despite the elevated concentrations of iron (6504 mg/kg) and chromium (341.03 mg/kg) detected in the soil, Pseudomonas stutzeri LBR exhibited consistent efficiency in mercury (Hg) bioremediation.

This ability is likely related to the strain's multimetal resistance profile. Previous studies have reported that P. stutzeri is resistant to chromium [1] (reference to be inserted) and capable of producing siderophores that bind iron, thereby reducing its bioavailability and potential toxicity [2] (reference to be inserted).

As noted by Nanda et al. (2019), bacterial strains tolerant to multiple metals are well suited for bioremediation purposes, as they tend to show better survival and functional performance in environments with complex metal contamination [3].

Here is the sentence included in the revised manuscript at Line 384:

“In the presence of elevated concentrations of iron (6,504 mg/kg) and chromium (341.03 mg/kg) in the soil, Pseudomonas stutzeri LBR maintained efficient Hg bioremediation, likely supported by its intrinsic multimetal resistance mechanisms [1,2]. The simultaneous presence of multiple background metals may promote bacterial adaptive responses, aligning the experimental setting more closely with the environmental complexity of real-world contaminated soils [3].”

1. Kumari D, Pan X, Zhang D, Zhao C, Al-Misned FA, Mortuza MGolam. Bioreduction of Hexavalent Chromium from Soil Column Leachate by Pseudomonas stutzeri. Bioremediation J. 2015;19: 249–258. doi:10.1080/10889868.2015.1029116

2. Essén SA, Johnsson A, Bylund D, Pedersen K, Lundström US. Siderophore Production by Pseudomonas stutzeri under Aerobic and Anaerobic Conditions. Appl Environ Microbiol. 2007;73: 5857–5864. doi:10.1128/AEM.00072-07

3. Nanda M, Kumar V, Sharma DK. Multimetal tolerance mechanisms in bacteria: The resistance strategies acquired by bacteria that can be exploited to ‘clean-up’ heavy metal contaminants from water. Aquat Toxicol. 2019;212: 1–10. doi:10.1016/j.aquatox.2019.04.011

3. Page 14, Lines 298–302: o Issue: Bacterial counts use "10º" (e.g., 27 x 10º CFU/mL) instead of "10⁹". o Correction: Replace "10º" with "10⁹" throughout.

Thank you for your comment. The correct notation (e.g., 10⁹) was already used in the manuscript, but it may have appeared unclear due to font size or formatting. We have increased the font size of the superscript to improve legibility throughout the text. (Line 296)

4. Page 16, Line 345: o Issue: Acknowledgments list collaborators but omit funding sources. o Recommendation: Declare funding or state "No funding was received."

Thank you for your comment. We have added the following sentence to the Funding section:“The authors received no specific funding for this work.” At line 474.

Data Availability (Throughout)

1.Raw data (bacterial counts, Hg measurements) are not deposited in a public repository.

Action: Add a Data Availability Statement (e.g., "Data are available in [repository name] at [DOI/link]").

The raw data supporting the findings of this study, including mercury concentrations and bacterial counts, have been deposited in the Zenodo repository under the title ‘Raw data on mercury concentrations and bacterial counts’, and are accessible via DOI: 10.5281/zenodo.16621459. Available online from August 30, 2025 (at line 478 in revised manuscript)

2. GenBank Accession (Page 6, Line 125)

The P. stutzeri LBR sequence is cited as deposited in GenBank, but the accession number is missing.

Action: Provide the GenBank accession number in the Methods.

Thank you for your comment. The GenBank accession number is already provided in the methods on page 8, line 178.

Waiting for your reply,

My thanks in advance,

Dr Chiraz ABBES

---

## [Editor Report · Decision Letter 2]

7 Aug 2025

Novel method for combining microbial bioremediation with static magnetic fields to remediate mercury-contaminated soils

PONE-D-24-56633R2

Dear Dr. Abbes,

We’re pleased to inform you that your manuscript has been judged scientifically suitable for publication and will be formally accepted for publication once it meets all outstanding technical requirements.

Kind regards,

Naga Raju Maddela, Ph.D

Academic Editor

PLOS ONE
---

## [Editor Report · Acceptance letter]

PONE-D-24-56633R2

PLOS ONE

Dear Dr. Abbes,

I'm pleased to inform you that your manuscript has been deemed suitable for publication in PLOS ONE. Congratulations! Your manuscript is now being handed over to our production team.

Kind regards,

on behalf of

Dr. Naga Raju Maddela

Academic Editor

PLOS ONE